# Optical Parameters for Using Visible-Wavelength Reflectance or Fluorescence Imaging to Detect Bird Excrements in Produce Fields

**Alan M. Lefcourt [1,*], Mark C. Siemens [2] and Paula Rivadeneira [3]**

[1] USDA (United States Department of Agriculture) Agricultural Research Service, Rm 21, Bldg 303, BARC-East, Powder Mill Rd., Beltsville, MD 20705, USA

[2] Department of Agricultural and Biosystems Engineering, University of Arizona, Yuma Agricultural Center, 6425 W. 8th St., Yuma, AZ 85364, USA; siemens@cals.arizona.edu

[3] Department of Soil, Water, and Environmental Science, University of Arizona Cooperative Extension, Yuma Agricultural Center, 6425 W. 8th St., Yuma, AZ 85364, USA; pkrivadeneira@email.arizona.edu

\* Correspondence: amlefcourt@gmail.com

**Abstract:** Consumption of produce contaminated with pathogens of fecal origin is the most common source of food borne illnesses. Current practice is to visually survey fields for evidence of fecal contamination, and to exclude problematic areas from harvest. Bird excrement is known to contain human pathogens, and is often not detectable in produce fields using current survey methods. The goal of this project was to identify parameters for optical detection of bird excrements to support development of instruments to be used to supplement existing visual surveys. Under daylight ambient conditions, results suggested that reflectance imaging at around 500–530 nm or 610–640 nm could be used to detect excrements from the three bird species tested. Images were acquired using ad hoc camera parameters; however, normalizing intensities for individual images at 525 nm and using a fixed detection threshold allowed detection of 100% of bird excrements with no false positives against the background that consisted of local soil and fresh romaine and spinach leaves. Similar results were obtained using fluorescence imaging. Fluorescent imaging was accomplished in a darkened room using 405-nm illumination. The largest consistent differences in intensity responses between excrements and the brightest non-excrement object in the background matrix occurred at around 550 nm. Results suggested that using reflectance or fluorescence imaging for detection of bird excrements could be a valuable tool for reducing risks of consuming contaminated produce. One possibility would be to incorporate appropriate reflectance imaging capabilities in drones under the control of the individuals currently conducting field surveys.

**Keywords:** birds; hyperspectral imaging; imaging; food safety; leafy greens; fecal matter

## 1. Introduction

Produce contaminated with fecal-derived pathogens is a recognized food-safety concern. Current practice is to visually survey fields prior to harvest for signs of fecal contamination, and to restrict harvesting of identified problem areas. The difficulty of detecting bird droppings in fields is a concern. To address this concern, this study examined optical parameters and methods that could be used to develop optical survey instruments to detect bird droppings in produce fields.

Fresh produce represents a particular food-safety risk as it is commonly consumed fresh, without an intervening kill step such as cooking to reduce any potential pathogen load. Painter et al. [1] reported that produce is the most common source of food borne illnesses, and incidences of produce-related food-safety events have been increasing in recent years [2]. Most incidents of non-viral foodborne

illness associated with consumption of produce can be traced to contamination of the produce in fields prior to harvest [3,4]. Thus, any action taken prior to or at harvest that can reduce the probability of contaminated produce entering the food chain will have the greatest potential impact on food safety. Steps taken in fields to alleviate contamination events include restricting access to fields, cleaning harvest equipment prior to and during harvest, and human hygiene requirements [5–7]. A critical step is the surveying of fields for fecal contamination. Field surveys at multiple production stages, including just prior to harvest, are required [5–7]. Generally, trained farm personnel walk through fields looking for indications of possible fecal contamination such as signs of animal intrusion including tracks, rooting, or crop damage. If a potential contamination site is identified, the site is flagged and the surrounding area is not harvested. If the signs of animal intrusion are extensive, food-safety personnel may flag the area on the assumption that the area is likely to contain fecal material. The problem with birds is that they can visit a field, leave their droppings, and depart without leaving any readily identifiable indications that they have ever been in the field. Birds, and particularly gulls, are known carriers of human pathogens [8–10], and have been implicated in the contamination of local waters [10–13].

The goal is to develop practical methods to allow fields to be surveyed for bird droppings. It would be impractical to ask personnel to visually examine each plant to fully survey the produce fields. Survey methods that can be automated or used without requiring personnel to enter fields are needed; it would be an added benefit if these methods could be used to detect fecal material from all types of animals. Our laboratory has developed numerous optical methods to detect fecal matters [14–22]. For field use, using fluorescence [14–17] or reflectance [18,22] imaging in the visible wavelengths are the most promising methods. Methods based on measuring fluorescence responses to UV or purple illumination are probably the most sensitive. Fluorescence methods have two principal drawbacks—it is very difficult or costly to detect fluorescence responses under ambient daylight conditions, and an illumination source is needed to elicit a fluorescence response. Acquiring reflectance images at two appropriate wavelengths can be used to detect cow feces; however, false positives are a problem [18]. It was hypothesized that the color of bird droppings might facilitate detection of bird droppings. Birds use only a single orifice for excretions, the cloaca; as a result, their droppings usually include a whitish component made of uric acid, a form of nitrogenous waste [23]. This white waste component may represent a target characteristic that could facilitate the detection of bird excrements. We recently demonstrated that dairy manure could be reliably detected using fluorescence imaging at dusk or at night [17]. Tests were conducted to determine if bird excrements could similarly be detected. Due to the need for a UV or purple illumination source, fluorescence methods would require use of a vehicle to transport any optical imaging system through fields. Reflectance imaging can be accomplished using ambient sunlight for illumination, and thus, there exists the possibility of using a drone to accomplish the imaging. An objective of this study was to determine how many and which reflectance wavebands could be used to detect bird droppings.

## 2. Materials and Methods

After locating and identifying local bird flocks in proximity to produce fields, excrement samples were collected from three species of birds using either trays of soil left in roosting areas or by transferring deposits to such trays. Fresh romaine and spinach leaves were placed on the trays near the deposits and images were acquired using a hyperspectral imaging system (Figure 1). Reflectance images were acquired under ambient outside conditions, and fluorescence images were acquired using 410-nm illumination after the trays were moved to a darkened room. Spectra for analyses were created using hyperspectral images and regions-of-interest within deposits, produce leaves, and soil. Spectra were contrasted to determine wavelengths where the difference in measured responses between excrements and the maximum response of produce or soil were consistently the greatest.

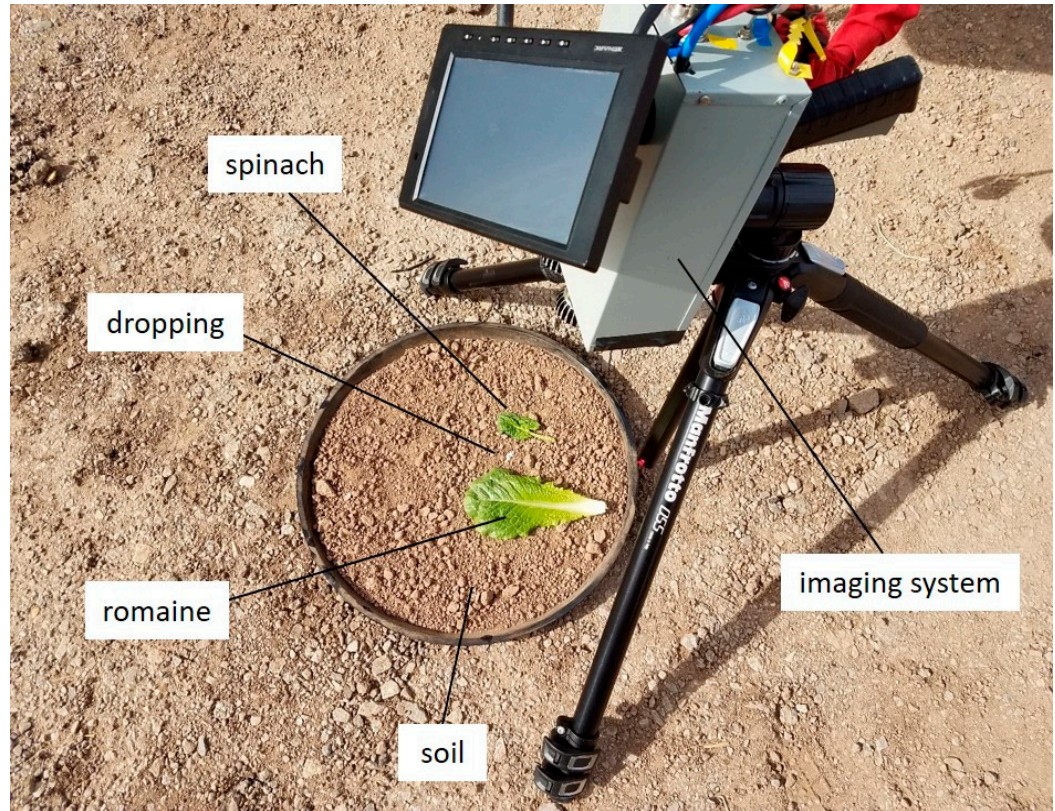

**Figure 1.** Experimental setup used to capture hyperspectral images of bird droppings against a background matrix that included a spinach leaf, a romaine leaf, and soil.

### 2.1. Hyperspectral Imaging System

The portable hyperspectral imaging system has been described in detail [19]. The system consisted of a C-mount lens (Rainbow S16mm, 2/3″, Sensei Shoj Co., Tokyo, Japan), a liquid–crystal tunable filter (400–720 nm, a 20-nm full width at half maximum (FWHM); VariSpec VIS, Caliper Life Sciences, Waltham, MA, USA), back-correction optics (Channel Systems Inc., Oakland, CA, USA), and a monochrome GigE CCD camera (Prosilica GC1380, Allied Vision Technologies, Stadtroda, Germany). Due to the narrow viewing angle of the tunable filter, the filter was placed between the lens and the camera, which necessitated the use of back-correction optics to allow images to be appropriately focused on the camera CCD. Illumination for fluorescence imaging was provided by four 10-watt, 405 nm, LEDs (LED Engin, Wellington, MA, USA). LEDs were mounted in aluminum finned casings to help dissipate heat and can be switched on or off. Power for illumination was provided by an 18-V lithium-ion battery pack. Power for other components (12 V) was provided by using a portion of the battery pack. Factory calibration of the tunable filter was checked using standards [19].

Image data were acquired using an X61 Thinkpad laptop and an acquisition program written in-house using Visual Basic version 6 (Microsoft, Seattle, WA, USA). The camera was connected to the laptop using the Ethernet port and the filter controller was connected using an USB port. The native resolution of the camera, 1360 (H) × 1024 (V) pixels, was binned by three to yield a final resolution of 453 × 341 pixels. Binning was accomplished to reduce noise, and to increase light sensitivity and dynamic range of acquisitions. The acquisition software facilitated the automated capture of a sequence of images from 465 to 720 nm at 5 nm increments. To acquire the sequence of images, the camera was mounted on a tripod so that the imaging field was identical for each of the sequential images that comprised the hyperspectral data set (Figure 1). Using a frame rate of 15 fps, a complete hyperspectral data set can be acquired in less than 30 s. The frame rate determined the maximum potential exposure

time for individual images. The 12-bit resolution of images was preserved by storing images using the 16-bit TIF grayscale format.

The camera supported gain settings from 1 to 32, and the acquisition software allowed individual gains to be set for each acquisition wavelength. For reflectance images, the camera gain was set to 1 at all wavelengths, the lowest gain possible. Aperture was selected by looking at continuously acquired images at a selected wavelength, and reducing the aperture until there was no saturation. For fluorescence images, the aperture was set to an f-stop of about 2.0, and gains were set to a fixed number across all wavelengths.

## 2.2. Sample Acquisition

In late February, 2018, the growing areas around the University of Arizona's Yuma Agricultural Center, Yuma, AZ were surveyed for bird flocks. Flocks of mourning doves (*Zenaida macroura*) and rock pigeons (*Columba livia*) were found in the Center. Flocks of ring-billed gulls (*Larus delawarensis*) were found in numerous nearby fields where romaine lettuce was being harvested. To attract birds to known roosting areas at the Yuma Agricultural Center, bird feed (Morning Song, Global Harvest Food, Ltd., Seattle, WA, USA) was laid out. To harvest excrement samples, trays containing local soil to a depth of about 10 mm were placed in areas of the Yuma Agricultural Center where birds congregated in the evening. Trays with droppings were collected the following day and imaged within 8 hours of collection. As the gulls were found on private properties, placing trays was problematic. Instead, both moist and dry droppings were transferred with the underlying soil using a shovel to trays already containing soil. These trays were imaged the same day droppings were collected.

To allow responses of droppings to be contrasted with responses of produce, fresh romaine and spinach leaves were placed on trays near droppings immediately prior to imaging (Figure 1). Fresh produce was obtained on imaging days from plots at the Yuma Agricultural Center.

## 2.3. Hyperspectral Data Acquisition

For reflectance images, data were acquired using a gain of 1. The wavelength was set to 525 nm and the aperture was adjusted so that the brightest area in the image was below saturation. A hyperspectral data series was then collected. As the sequential images were being acquired, the images were visually monitored for saturation. If saturation was seen, after completion of acquisition of the hyperspectral data series, the aperture was reduced and another acquisition was initiated. For fluorescence imaging, the aperture was set to an f-stop of about 2. The wavelength was set to 525 nm, and the gain was adjusted so that the brightest area in the image was below saturation. A hyperspectral data series was collected using this gain. In a procedure similar to that used for reflectance imaging, if saturation was noted at any wavelength, the gain was reduced and a new data series was collected. A large number of data series were collected is this manner. These procedures allowed the exposure time to be maximized.

Preliminary analyses indicated the hypothesis that the white areas of bird droppings would facilitate the detection of bird dropping was correct, and it would be relatively easy to detect bird droppings against a background matrix that included soil and plant materials. Considering this finding, three hyperspectral data series for each of the three bird species were selected for a detailed analysis.

## 2.4. Analysis

Image data were analyzed using a software program written in-house using Visual Basic version 6 (Microsoft, Seattle, WA, USA). First, the acquired sequence of TIF image files for a single hyperspectral data set was transformed into the ENVI (Harris Co., Melbourne, FL, USA) file format. For ENVI files, the program allowed visualization of images at any selected wavelength. Any image within the data set can be used to define a region-of-interest (ROI) that is carried across all wavelengths, and multiple ROI can be defined for each data set. A single circular ROI was used for a selected bird dropping. The size of the circle was selected to be as large as possible while still including only bird

excrement. A similar-sized circle was placed on a bright area on the romaine and spinach leaves, and on the brightest area of soil. Spectral intensities were averaged across each of these four ROI at each wavelength, and averages for each ROI were displayed on a single graph of average intensity by wavelength. Graph data were transferred to spreadsheet files to allow composition of drawings for publication. The spectral graphs were used to determine spectral characteristics that might allow automated differentiation of targets from background and possible confounding objects. To validate the ability to detect bird droppings across the nine hyperspectral data sets chosen for detailed analyses, images at 525 nm were normalized [20] and subjected to a threshold filter. As discussed in the Results and Discussion Section, the rib areas of some romaine lettuce leaves were masked-out prior to detection.

## 3. Results and Discussion

### 3.1. Goals

The goal of this project was to determine if it might be feasible to use spectral imaging to detect bird excrement in produce fields. Prior work had demonstrated that using reflectance spectral imaging to detect fecal materials from animals such as cows was possible with a high rate of detection confounded by some specific classes of false positives [18], and that fluorescence spectral imaging could achieve 100% detection with no false positives when images were acquired during dusk or evening [17]. Birds have a unique method of dealing with body wastes. They convert ammonia wastes to uric acid rather than urea, reducing their water consumption needs; the uric acid combines with their feces prior to excretion. Uric acid is the predominant component in bird droppings and is responsible for the white or cream-colored aspect of droppings [23]. It was hypothesized that this colorization would facilitate detection of bird droppings using imaging techniques. The study was a survey study and was not designed to test the validity of any specific detection algorithm. Validity testing would best be accomplished using instrumentation appropriate for use in a commercial system. To address concerns that laboratory studies do not necessarily reflect real world conditions, samples were collected from wild bird populations in their native habitats and, where feasible, measurements were made outdoors under ambient environmental conditions.

### 3.2. Using Spectra for Detection

For a detection system to be commercially viable, cost is an overriding consideration. Cost considerations suggest that the optical system use at most one or two wavelengths for imaging and that the system include only limited computational capabilities. Cost considerations also require essentially no false positives due to the potential expense of the unwarranted condemnation of a large area of a produce field. Thus, only simple, robust, detection schemes needed to be considered. Fortunately, results demonstrated that bird droppings could be readily detected against a background including soil and plant materials using images acquired at a single wavelength and a threshold filter. As can be seen, this detection scheme is robust because it could utilize any of the wide range of wavelengths. Because of the simplicity of the detection scheme, the use of statistical methods to identify appropriate wavelengths is not necessary. Furthermore, an attempt to statistically validate the detection scheme is not warranted as the equipment used in this study is not appropriate for commercial use. Field studies of the validity of the detection scheme should use a cost effective optical system that includes a monochrome camera and a single wavelength interference filter.

Previous study results suggested that for fluorescence imaging, 525 nm would be a suitable wavelength for the detection of dairy manure against a background matrix that included produce and soil [17]. Given this finding, measured responses at 525 nm were used to test system settings for saturation prior to initiating acquisition of hyperspectral data sets. Fluorescence spectra for the three species tested are somewhat dissimilar; representative spectra are shown in Figures 2–4. However, all species showed broad elevated responses that peaked around 640 nm for dove and around 550 nm for

pigeon and gulls. Responses for produce were minimal below 650 nm, and responses for soil were minimal at all wavelengths. Some individual dirt clods did show low levels of response in the blue and green regions that were well below the relative response intensities of bird droppings. The response peak at around 635 nm for excrement in Figure 2 was unique across tested droppings for all bird species, as well for over ten additional dove droppings that were not included in the data sets selected for detailed analyses. It was assumed that this peak was due to something unusual in that bird's diet. Figure 2 was selected for presentation as an example of the need to examine a large sampling of droppings to fully elucidate potential confounding factors. Produce showed the expected peak due to chlorophyll and its by-products at around 685 nm [24]. However, the magnitudes of the peak varied from leaf to leaf, with a tendency for a greater magnitude of response from romaine leaves compared to spinach leaves. It was also noticed that responses seemed to be greater in fresh leaves compared to leaves harvested earlier in the day. An implication of this finding was that fluorescence studies that used store-bought produce may underestimate responses that might be encountered in field measurement situations.

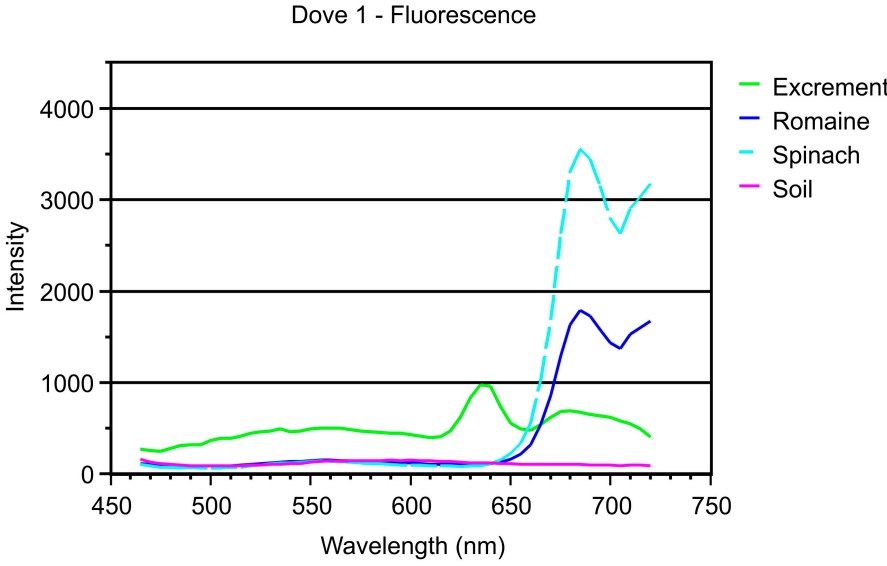

**Figure 2.** Fluorescence spectra for a dove dropping, romaine leaf, spinach leaf, and soil.

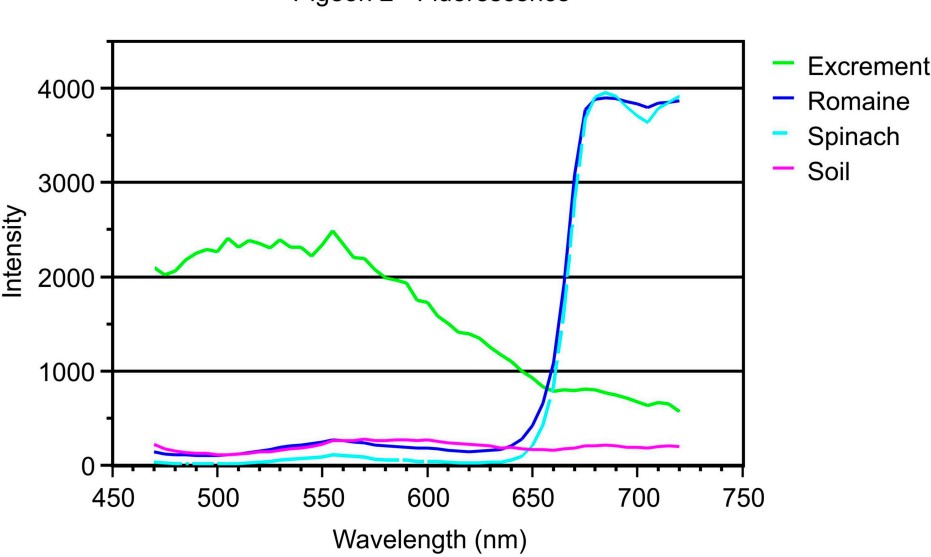

**Figure 3.** Fluorescence spectra for a pigeon dropping, romaine leaf, spinach leaf, and soil. Note that data from the image at 465 nm were excluded as the image was found to have been corrupted.

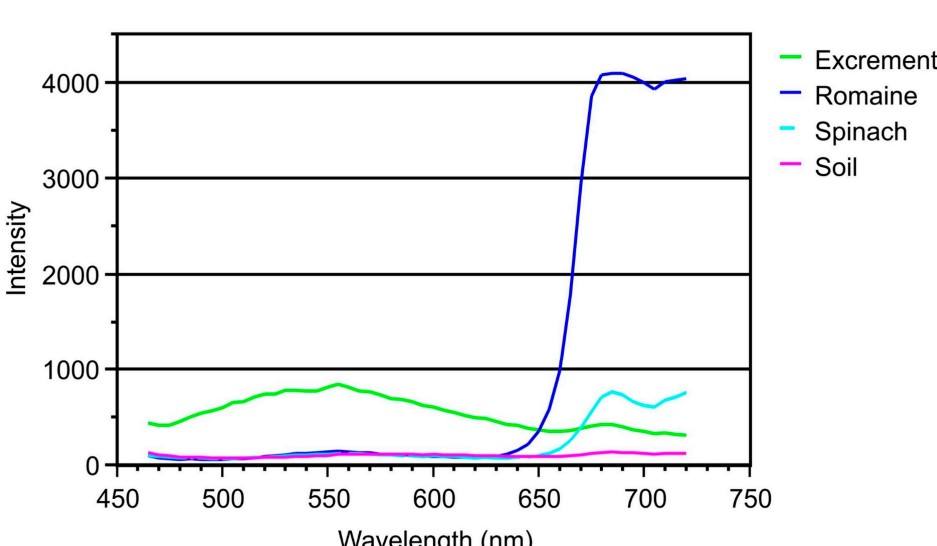

**Figure 4.** Fluorescence spectra for a gull dropping, romaine leaf, spinach leaf, and soil.

As mentioned, a prior study determined that 525 nm was a suitable wavelength for detecting dairy manure using fluorescent imaging, and dairy manure had similar fluorescence response characteristics as manure from other 4-legged animals [24]. The 525 nm responses for birds are close in magnitude to peak responses; thus, 525 nm is probably a good choice for designing optical instrumentation to use fluorescence to detect all types of animal excreta in produce fields. Figure 5 shows the raw image of a gull dropping at 525 nm and the same image scaled to emphasize a lower intensity response to allow the produce leaves to be identified in the image. The 525-nm image is for the same gull dropping used to produce the spectra in Figure 4.

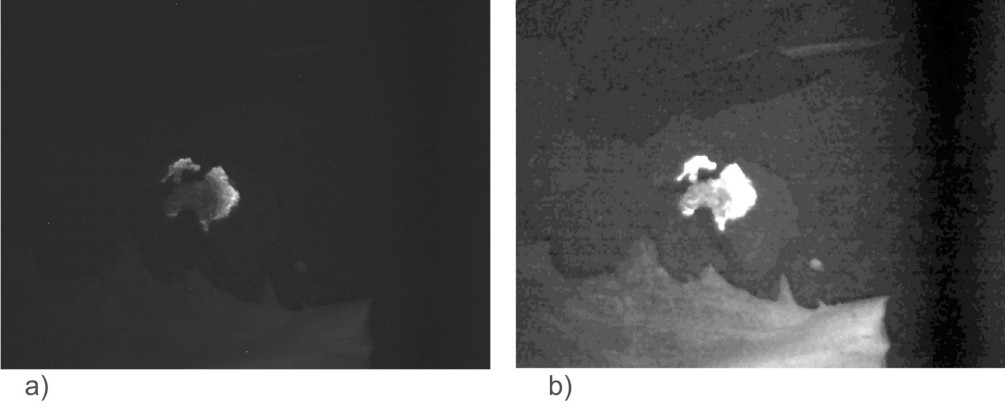

a)        b)

**Figure 5.** Fluorescence images of gull dropping, romaine leaf, spinach leaf and soil at 525 nm. Image (**a**) is the raw image, and image (**b**) is the image filtered using intensity scaling to emphasize low intensity aspects of the image so that the spinach leaf (above) and romaine leaf (below) are more easily visible.

The patterns of measured reflectance spectra were similar across bird species (Figures 6–8). For bird excrement, ROI were selected that encompassed the brightest areas of droppings; in general, spectral intensities increased with increasing wavelength. Spectra for soil showed a similar pattern to droppings, but at lower relative intensities. Spectra for romaine and spinach were similar in pattern with a peak around 560 nm, a valley around 640–670 nm, and a sharp rise starting at around 650–680 nm. For all droppings, the measured intensities for droppings exceeded the intensities of soil and plant materials for wavelengths below 640 nm. Regions where intensities were high for bird droppings and where relative differences from soil and plant materials were consistently greatest

were around 500–530 nm and 610–640 nm. In reality, any wavelength 640 nm or less could be used to discriminate bird excrement from the background matrix.

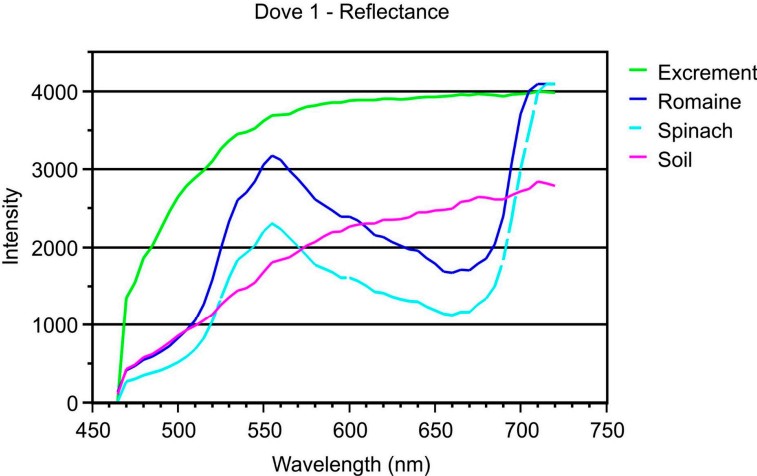

**Figure 6.** Reflectance spectra for a dove dropping, romaine leaf, spinach leaf, and soil.

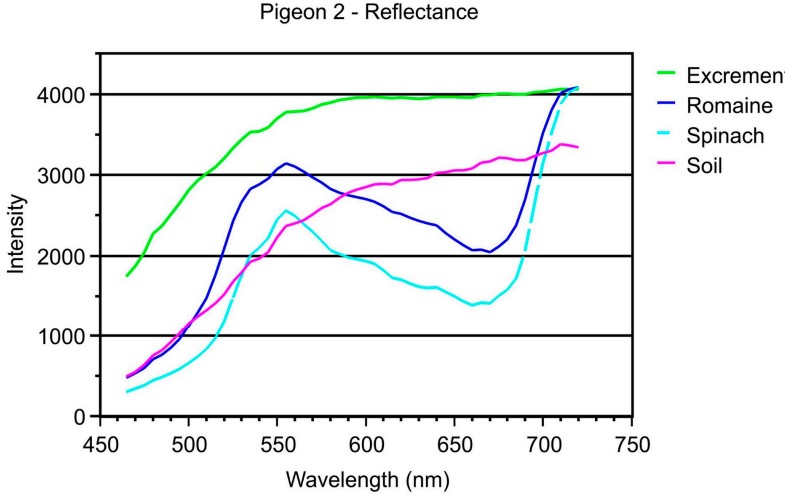

**Figure 7.** Reflectance spectra for a pigeon dropping, romaine leaf, spinach leaf, and soil.

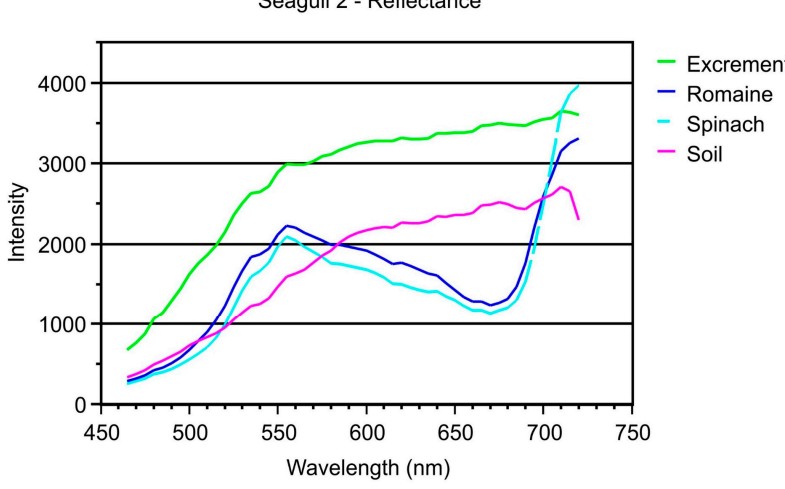

**Figure 8.** Reflectance spectra for a gull dropping, romaine leaf, spinach leaf, and soil.

In a prior study examining the detection of dairy cow feces on spinach leaves, images at 690 nm and 710 nm were used to construct ratio images with a threshold used for detection [18,22]. Bird droppings contain both urine and feces components; the urine component results in the "white" aspect of droppings and the feces is responsible for the "dark" aspect of droppings. The spectra for droppings shown in Figures 5–7 are for white areas. In many of the samples of droppings, there were little or no dark areas; thus, attempting to use the prior findings for feces as a basis for detection of bird droppings was not feasible. Furthermore, the measured intensities for romaine and spinach at both 690 nm and 710 nm often showed maximums similar to those measured in white areas of droppings. Thus, to be able to reliably detect both bird droppings and feces from other animals using reflectance imaging would require taking images at the 690 nm and 710 nm wavelengths used in the prior study and, as results of this study suggest, at a third wavelength below 640 nm. Alternatively, it might be feasible to use two wavelengths. Examination of the principal component data presented in the background study that was used to select the 690 and 710 nm wavelengths suggested that it might be feasible to replace the 690-nm wavelength with a wavelength around 650 nm [16]. The 650-nm wavelength would be suboptimal for detecting both droppings and feces; however, further testing might demonstrate that this wavelength would be adequate for reliably detecting both. Unfortunately, the acquired reflectance images in the current study in the red region often had areas of saturation that made formal testing of this hypothesis using data untenable. However, the potential difficulty of using 650 nm to detect bird droppings was evident for one of the gull sample data sets. Figure 9 shows reflectance images at both 525 nm and 650 nm. At 525 nm, a simple threshold filter could be used to discriminate the dropping. At 650 nm, the brightest area of the dropping and the dirt clod were both saturated; however, the relative extent of the area of saturation for the dropping suggested that the dropping was brighter than the dirt clod. In any case, if the images were being reviewed by a human observer, the observer should easily be able to distinguish between the dirt clod and the dropping. This 650-nm image had one particularity that impacted detection under ambient conditions—there was a shadow passing across the image with the dropping in the shadow while the dirt clod had full sun exposure. This occurrence emphasized the potential difficulty of using measurements taken under real world conditions for detection.

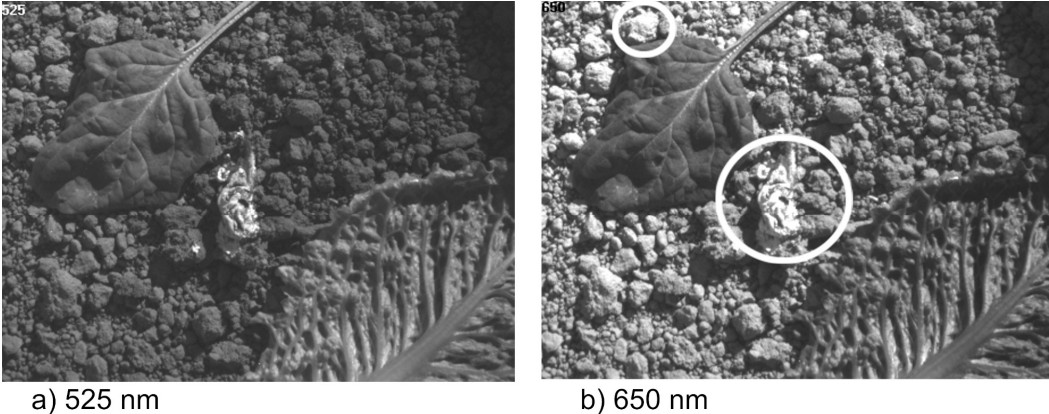

a) 525 nm          b) 650 nm

**Figure 9.** Reflectance images of a gull dropping at (**a**) 525 nm and (**b**) 650 nm. Note that the brightest areas of the circled dirt clod and dropping in the 650-nm image were both saturated; however, the larger relative extent of the saturated area for the dropping suggested that the dropping was brighter at its maximum than the dirt clod.

Spectral characteristics are not the only factors that should be considered when designing an optical instrument to survey fields for bird droppings. Other factors to consider include size of bird droppings, actual time birds spend in fields, excretion traits, and probability of carrying pathogenic bacteria. Dove droppings are small, some 2–4 mm in diameter, and identifying the smaller droppings in low resolution images would be difficult, if not impossible. Both the doves and the pigeons were

found to be roosting outside of fields. Gull flocks were found in fields, commonly on the ground in open areas next to irrigation canals. Gulls were observed to excrete on take-off and were also observed to take-off over irrigation canals. Numerous droppings were found immediately next to the canals, and it was reasonable to assume some droppings landed in the canals. Gulls are known carriers of pathogenic *E coli.* [8–10], and migrating gulls have been linked to increases in E. coli in municipal water sources [10–13]. When this study was conducted, numerous flocks of gulls were seen. Per locals, the high level of gull presence was a recent phenomenon. The flocks were found almost exclusively in fields of romaine lettuce that were being harvested, which was an interesting observation given the *E. coli* outbreak linked to consumption of romaine lettuce harvested in Yuma in March of 2018 [25]. More study is needed of birds as potential vectors of pathogen contamination of produce; however, existing knowledge concerning gulls suggests that development of optical instrumentation to detect gull droppings in particular would be warranted.

### 3.3. Practical Aspects of Detection

Data acquisition parameters for this study were selected to allow comparison of relative responses across all wavelengths of a given hyperspectral data set and not to optimize acquisition parameters for detection at any particular wavelength. In this regard, for reflectance imaging, exposure time was maximized and aperture was adjusted to eliminate saturation at the worst-case wavelength across the acquired spectrum. Furthermore, the aperture correction was crude as it was done visually and not by image analysis. The stated conclusion of this study was that detection of bird excrements could be accomplished using any of the large range of wavelengths and suggested the use of 525 nm. However, the acquisition protocol resulted in images at 525 nm that were inconsistent in intensity across hyperspectral image sets. To address this inconsistency, the 525-nm images were normalized based on intensity histograms. An alternate method commonly used to normalize the effects of differences in illumination intensity is to use a ratio of images acquired at two different wavelengths. However, ratio calculations would require additional computations and it was likely, given the dramatic differences in relative spectral responses seen in this study, that normalization would not be necessary if image acquisition parameters were optimized for 525-nm images and for actual ambient illumination. Another problem that was encountered related to the color of some of the ribs of romaine leaves. Presumably due to lack of exposure to the sun, the problem ribs had little pigmentation. In commercial fields prior to harvest, the problem ribs would not have been visible. The problem rib areas were masked-out prior to detection. Using this detection scheme, bird droppings for all of the nine hyperspectral data sets selected for in-depth analyses were detectable using a fixed threshold of 3000. There were no false positives. For fluorescence detection, using the raw images, 100% of bird droppings could be detected at 525 nm using a threshold of 250 with no false positives. These examples are not meant to suggest use of a specific detection scheme, but only to emphasize the robustness of the ability to detect bird droppings using imaging techniques.

For both reflectance and fluorescence imaging, because of the ease of differentiating bird droppings from the background matrix across a wide range of wavelengths, selection of the most appropriate wavelength and bandwidth for imaging will depend more on equipment considerations and less on the actual wavelength used for detection. Critical consideration would involve questions such as the cost of components verses overall sensitivity. The least expensive components that might be used include a monochrome camera and a single-wavelength interference filter. Cost considerations for the camera would revolve around factors such as sensitivity, noise, resolution, weight, and dynamic range. A high resolution camera might be able to detect small bird droppings such as those seen for mourning doves. For the filter, there are generally two types of interference filters. The less expensive filters have a bell-shaped transmittance distribution across wavelengths. These filters are described in terms of the center frequency and a pass-band where the edges of the pass-band are the wavelengths where the transmission is half the maximum at the center (FHHM, full height at half max). The downsides of this type of filter are that significant light is transmitted at wavelengths outside the pass-band, and the

transmittance at the center wavelength can be low (commonly less than 50% in the blue-green region). The more expensive filters have a pulse-shaped transmittance distribution with sharp edges and a flat top. The advantages of this type of filter are that little light is transmitted outside the pass-band, and that the transmittance across the pass-band approaches 100%. The disadvantage is cost. The final selection of equipment will also depend on implementation. For example, for use with a drone, weight of components is a critical consideration.

*3.4. Implications*

The results indicate that fluorescence responses to UV illumination could be a sensitive method for detecting bird droppings. The primary disadvantages of fluorescence imaging are that an illumination source is required and imaging under practical conditions would have to be done under low ambient light conditions. The advantage is that images could be acquired at a single wavelength with 525 nm being a good wavelength to consider for use. Surveying fields at night is reasonable given that night operations are already common. The 525-nm wavelength would also be a good candidate for using reflectance imaging to detect bird droppings. A particular advantage of reflectance imaging is that the sun can provide illumination, which allows the imaging system to be mounted on some sort of aerial drone. The drone could be used as a supplement to current survey practices. If the goal of using reflectance imaging were to detect both bird droppings and other animal feces, the situation would become more complicated as discussed above. Considering the most likely failing of current survey methods relates to the difficulty of detecting bird droppings in fields, development of a drone that uses 525-nm reflectance imaging and that can be readily deployed as part of surveys should be a priority.

**Author Contributions:** A.L. conceived of this project, conducted the experimental trials, and wrote the manuscript. M.S. helped conduct the trials, reviewed the manuscript, and provided background information on practical considerations for use of the optics in commercial fields. P.R. located the bird flocks, established feeding/sampling areas for birds, and reviewed the manuscript.

**Acknowledgments:** This work was funded by the United States Department of Agriculture (USDA). Facilities were provided by the University of Arizona. The USDA is an equal opportunity employer.

**Conflicts of Interest:** The authors declare no conflict of interest.

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
