# Peer review of "Optical Parameters for Using Visible-Wavelength Reflectance or Fluorescence Imaging to Detect Bird Excrements in Produce Fields"

_applsci, doi:10.3390/app9040715_

Round 1
Reviewer 1 Report
The paper concerns to the feasibility to use a portable hyperspectral system to detect faecal contaminations on fresh produce.
General comment:
There experiments are well designed and the procedure to acquire and analyze the spectra are good explained.
However, there are some outbreaks to fix in order to consider the paper for publication.
The main point is the absence of the statistical analysis on the spectra obtained. Moreover, the number of samples analyzed was missed in the text.
Below, some specific comments.
Specific comments:
Line 64 -70: Please improve the reference citations for the Hyperspectral techniques employed in the field; these papers could give to you a support to valorize your thesis. I suggest to you these one:
· Everard, C. D., Kim, M. S., & O'Donnell, C. P. (2016). Distinguishing bovine fecal matter on spinach leaves using field spectroscopy. Applied Sciences (Switzerland), 6(9) doi:10.3390/app6090246
· Everard, C. D., Kim, M. S., Cho, H., & O’Donnell, C. P. (2016). Hyperspectral fluorescence imaging using violet LEDs as excitation sources for fecal matter contaminate identification on spinach leaves. Journal of Food Measurement and Characterization, 10(1), 56-63. doi:10.1007/s11694-015-9276-x
· Romaniello, R., Peri, G., Leone, A. (2016).Fluorescence hyper-spectral imaging to detecting faecal contamination on fresh tomatoes. Journal of Agricultural Engineering, 47 (1), pp. 7-11.
· Everard, C. D., Kim, M. S., Siemens, M. C., Cho, H., Lefcourt, A., & O'Donnell, C. P. (2018). A multispectral imaging system using solar illumination to distinguish faecal matter on leafy greens and soils. Biosystems Engineering, 171, 258-264. doi:10.1016/j.biosystemseng.2018.05.001
Line 105-107. Indicate please the spatial and spectral resolutions.
Line 113-118. The sentence is not completely clear. Did you use the automatic gain adjustment? If not, please delete the first sentence.
Figure 1-3. Please explain the variation of intensity of “Romanie” in fig 1, compared to fig 2 and 3; the statistical analysis is missed.
Figure 4-7. As figure, 1-3. The statistics are missed. Please indicate the significance on the selected wavelengths used for discriminations.
Author Response
Thank you for your reviewing. Please see attach.

Reviewer 2 Report
Manuscript ID: applsci-404581
Title: Optical parameters for using VIS reflectance or fluorescence imaging to detect bird excrements in produce fields.
The present manuscript describes a very preliminary study that shows the possible feasibility of the identification of fecal waste in vegetables. The objectives and approach presented are really interesting. However, the data obtained can be studied in more depth and, therefore, a more exhaustive discussion can be provided. The work seems to me an incomplete study. In my view, Material and methods and Results and discussion sections should be extended with some statistical methods for testing the feasibility of using VIS reflectance or fluorescence imaging to detect bird excrements in produce fields. For that, I suggest a major revision of this manuscript before publication. Moreover, the following points should be clarified:
Lines 64-70: Have you tried to get a general method for the detection of all these animal feces? Or different environmental conditions?
Lines 82-90: Could you add an image or scheme to show how the samples were arranged and how the images were acquired?
Lines 108-109: Thus, your device does a wavelength scan and not a spatial scan, doesn’t it?
Line 118: Have you developed some image or spectra calibration? Two point calibration for example?
Lines 132-134: I can not clearly understand what the arrangement of the sample was like ... I insist on the addition of an image or scheme. Moreover, I think that it would be more appropriate to use leaves with the feces on them or stained with feces. They will represent better the feces contamination in produce.
Lines 136-137: Have you published this software anywhere? Are there some publications or property registrations to cite in this part of your manuscript?
Lines 146-148: Did you use some discrimination tool (LDA, DPLS, PCA, etc) for doing this? If yes, please, describe it.
Lines 160-161: Do you know if NIR region has been used with similar purposes? Add some reference if possible.
Figures 1-3: You might merge these three graphs in only one.
Figure 1: This graph marked with the change management assistant of the text editor.
Figure 4: These figures are really hard to see and interpret.
Line 233: Did you carry out a principal component analysis? Why is not PCA mentioned above? You need describe it in M&M section.
Lines 253-254: If the signal is saturated, we can not know what area is brighter...
The work seems to me an incomplete study. In my view, Material and methods and Results and discussion sections should be extended with some statistical methods for testing the feasibility of using VIS reflectance or fluorescence imaging to detect bird excrements in produce fields.
Author Response

(The authors gave the same response as above.)

Round 2
Reviewer 2 Report
Since authors do not carriedo the main querries I suggested, I can not change my recommendation. Major revission.
Author Response
The comments from the first round of review were helpful. There were few
comments from the second round. No comments were posted from Reviewer
1, so we assume we adequately addressed their concerns. Reviewer 2
raised their formal ratings in three categories, but provided no
specific recommendations.
In the first revision, we provided a detailed argument in Section 3.2 as
to why use of formal statistics would be inappropriate for the data in
this study. The section was edited for improved clarity. Further,
details of the number of samples, number of hyperspectral data sets and
sampling protocol were added in Sections 2.3 and 2.4. We feel that these
revisions, along with the explanatory statement in Section 3.1 that
“The study was a survey study and was not designed to test the validity
of any specific detection algorithm.” addresses the initial concerns
raised by reviewers about the use of statistics.
To address the concern that the spectra presented in the manuscript are
not representative of all the data collected, text was added in Sections
2.4 and 3.3. In brief, images at 525 nm were used to detect bird
droppings. Separate analyses for reflectance and for fluorescence images
detected 100% of droppings with no false positives.
With these revisions, we feel we have addressed all of the reviewers’ concerns.
Round 3
Reviewer 2 Report
I have already recommended doing a major revision of the manscript twice. However, the authors did not carry out any of my requests. In the second revision, I asked that the requests made in the first revision be carried out. I know that sometimes it is impossible to develop new analytical assays or collect more samples, However, I do not understand why the authors do not develop more chemometric procedures to complete the study. That is why I suggest rejecting this manuscript.